# Few-Shot Multimodal Explanation for Visual Question Answering

## ABSTRACT

A key object in eXplainable Artificial Intelligence (XAI) is to create intelligent systems capable of reasoning and explaining real-world data to facilitate reliable decision-making. Recent studies have acknowledged the importance of providing user-friendly and verifiable explanations to facilitate trustworthy Visual Question Answering (VQA) systems. This paper aims to promote explainable VQA from both data and method perspectives. First, we propose a new Standard Multimodal Explanation (SME) dataset and a new Few-Shot Multimodal Explanation for VQA (FS-MEVQA) task, which aims to generate the multimodal explanation of the underlying reasoning process for solving visual questions with few training samples. Our SME dataset includes 1,028,230 samples composed of questions, images, answers, and multimodal explanations, which can facilitate the research in both traditional MEVQA and FS-MEVQA. To the best of our knowledge, this is the first large-scale dataset with joint language-vision explanations based on standard English and additional visual grounding tokens, which bridge MEVQA to a broad field in Natural Language Processing (NLP). Second, we propose a training-free Multimodal Explaining Agent (MEAgent) method based on an LLM agent with multimodal open-world tools to infer answers and generate multimodal explanations for visual questions. Our MEAgent can learn multimodal explaining from merely $N(=16)$ training samples and leverage open-world abilities to perform FS-MEVQA on test samples. Comprehensive experimental results evaluated by language quality metrics, visual detection metric, and visual attribution metrics on our SME dataset indicate the superiority of our method for FS-MEVQA, compared to state-of-the-art MEVQA methods and the multimodal LLM GPT-4V. Our code and data are available at https://anonymous.4open.science/r/FS-MEVQA-646D/.

## CCS CONCEPTS

• **Computing methodologies → Knowledge representation and reasoning**; *Computer vision*; *Natural language processing*.

## KEYWORDS

Few-shot learning, multimodal generation, explainable artificial intelligence, visual question answering

**ACM Reference Format:**
Anonymous Authors. 2024. Few-Shot Multimodal Explanation for Visual Question Answering. In *Proceedings of the 32nd ACM International Conference on Multimedia (MM'24), October 28-November 1, 2024, Melbourne,*

**Unpublished working draft. Not for distribution.**

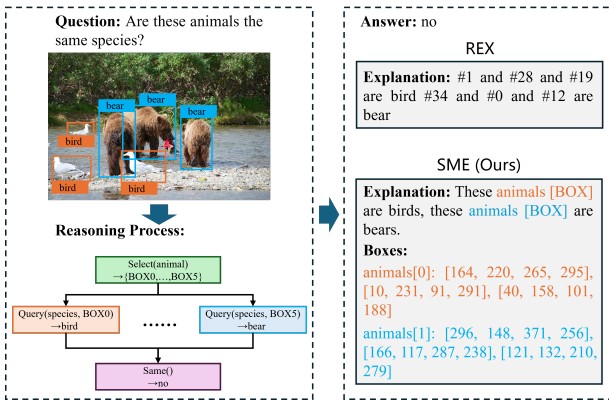

**Figure 1: An example of multimodal explanations constructed in the REX dataset [7] and our SME dataset. In REX, *#i* denotes boxes pre-extracted by Faster R-CNN, belonging to only 81 classes. We use [*BOX*] to represent the detection boxes needed in the explanation and annotate these boxes with their corresponding names mentioned in the explanation, based on the scene graph annotated by humans. Our explanation is in standard English with additional [*BOX*].**

*Australia.* ACM, New York, NY, USA, 10 pages. https://doi.org/10.1145/nnnnnnn.nnnnnnn

## 1 INTRODUCTION

Reasoning is a fundamental element of human intelligence and a crucial challenge in artificial intelligence [9, 10, 18]. Traditional reasoning models typically provide answers without offering explanations for their reasoning. This limitation greatly restricts their applicability, particularly in safety-sensitive situations, such as healthcare, transportation, and finance. To address this problem, a key object in eXplainable Artificial Intelligence (XAI) is to create intelligent systems capable of reasoning and explaining real-world data to facilitate reliable decision-making [11, 37, 42]. The recent development of Large Language Models (LLMs) has led to remarkable progress in reasoning on textual data and generating textual explanations for reasoning processes [12, 26]. However, when it comes to the field of multimodal reasoning, tasks necessitate the ability to comprehend multimodal content and generate multimodal explanations to reveal the underlying reasoning processes, which is still a challenging problem.

While Visual Question Answering (VQA) [3, 4, 13, 15, 32] is a typical and important multimodal reasoning task, some pioneering work has been made in Multimodal Explanation for VQA (MEVQA) [7, 50, 51]. Zellers [51] construct the VCR dataset with four explanation choices for every question. The proposed task is to select the correct explanation, which is usually impractical. Moreover, VCR only considers people as visual objects in explanation, limiting its application range. Recently, Chen and Zhao [7] convert the reasoning step graph to multimodal explanations, proposing the REX dataset and a generative task, as shown in Figure 1. However,

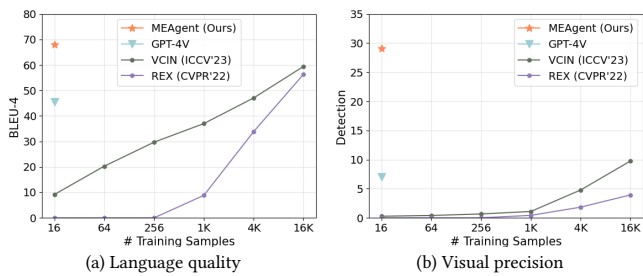

(a) Language quality    (b) Visual precision

**Figure 2: Performance of MEVQA methods with varying number of training samples.**

their customized format reduces the readability of the explanations. Moreover, they use a Faster R-CNN [36] trained on MS-COCO [25] to extract object boxes for explanation annotation, which can only detect 81 classes and bring inaccuracy in explanations. Furthermore, existing generative methods for multimodal explaining [7, 50] rely on large-scale training data, which is costly and may be impractical in various vertical domains. As shown in Figure 2, when trained on a small amount of data, state-of-the-art methods experience a notable decline in performance, despite being built upon pretrained Vision-Language Models (VLMs). Especially considering the recent progress in open-world learning (e.g., LLMs), more streamlined and applicable learning frameworks may represent the future direction. Therefore, we propose the Few-Shot MEVQA (FS-MEVQA) task that aims to learn MEVQA from few training samples. Meanwhile, GPT-4V, which may currently be the most powerful multimodal LLM, also performs unsatisfactorily, showing the flaws of LLMs in explaining the implicit multimodal reasoning processes. Inspired by the observation of the above pioneering works in MEVQA, we attempt to promote MEVQA from both data and model perspectives, proposing a new multimodal explanation dataset and a novel LLM agent-based training-free method for FS-MEVQA.

To improve the readability and user-friendliness of multimodal explanations, we propose the Standard Multimodal Explanation (SME) dataset, where the explanations are in standard English with additional [BOX] tokens for visual grounding, as shown in Figure 1. Our constructed explanations are based on reasoning steps for solving the visual questions, with the associated key objects in Visual Genome scene graphs [17] of 1,703 classes. After converting the structured reasoning steps into natural language-like explanations with visual object annotations, we utilize GPT-3.5 [6] to revise the explanations without [BOX] into standard English. Moreover, to facilitate an effective evaluation of visual objects grounded in multimodal explanations, we annotate key visual objects with their corresponding names mentioned in explanations and add special [BOX] tokens to represent grounded visual boxes. We further propose a visual detection metric to evaluate the ability to simultaneously generate names and ground regions of key visual objects to explain the multimodal reasoning processes. As shown in Table 1, compared to the existing explanations for VQA, our multimodal explanations based on standard English with additional [BOX] tokens can be more user-friendly and expressive with far more visual objects involved in explanation, bridging MEVQA to a broad area in Natural Language Processing (NLP).

To overcome the dependency on large-scale training data, we propose a training-free Multimodal Explaining Agent (MEAgent)

**Table 1: Comparison of explanations for VQA. *Visual Object* denotes visual objects grounded in explanation.**

| Dataset | Modality | Visual Object | Format |
|---|---|---|---|
| VQA-E [22] | Language | None | Standard English |
| VCR [51] | Language-Vision | Only people | Customized |
| REX [7] | Language-Vision | 81 | Customized |
| SME (Ours) | Language-Vision | 1,703 | Standard English with [BOX] |

method based on an LLM agent with multimodal open-world tools to infer answers and generate multimodal explanations for visual questions, given merely $N(= 16)$ training samples. Traditionally, $N-shot$ few-shot learning represents given $N$ training samples for every class (i.e., every answer in VQA) [8, 39, 45], which is still costly and cannot address unseen classes in the test. Moreover, considering the recent progress in open-world learning [16, 19, 31, 35, 49], the paradigm of learning reasoning knowledge for every class may be outdated. Therefore, we propose a stronger and more practical few-shot learning setting for MEVQA, where only $N$ training samples are given for all classes. Under such a setting, the value of training samples mostly lies in defining the MEVQA task, while the model needs out-of-training knowledge and open-world tools to solve the test questions. Inspired by the recent work in LLM-based visual programming [14], we construct a GPT-3.5-based LLM agent with multimodal open-world tools, such as an open-world object detector, image croppers, and customized Python functions. With a few-shot program prompt and a few-shot explanation prompt constructed based on $N(= 16)$ in-context examples, our MEAgent can generate the multimodal program for solving the input question, execute it via open-world tools to infer the answer, and translate the execution process into a multimodal explanation of the multimodal reasoning process. Additionally, we propose a rethinking mechanism to complete the visual objects needed in the explanation but ignored in multimodal programming. Extensive experiments demonstrate that MEAgent significantly outperforms the state-of-the-art MEVQA methods [7, 50] with even thousands of training samples and the multimodal LLM GPT-4V [1] for FS-MEVQA.

In brief, the contributions of our work are as follows:

- We propose SME, a new dataset for Multimodal Explanation for Visual Question Answering (MEVQA) comprising 1,028,230 samples, with 1,703 visual objects requiring detection in explanations. To our knowledge, this is the first dataset where the explanations are in standard English with additional visual grounding tokens, bridging MEVQA with a broad area in NLP.

- We propose MEAgent, a novel training-free method based on an LLM agent with multimodal open-world tools for Few-Shot MEVQA (FS-MEVQA). MEAgent can infer answers and generate multimodal explanations of the reasoning processes for visual questions, given only $N(= 16)$ training samples. Additionally, we propose a rethinking mechanism to complete the visual objects needed in explanation but ignored in multimodal programming.

- Extensive experiments demonstrate the effectiveness of our dataset and method for FS-MEVQA. We adopt language quality metrics, visual detection metric, and visual attribution metrics to evaluate the generated multimodal explanations. Experimental results show the superiority of our MEAgent compared to the state-of-the-art MEVQA methods and the multimodal LLM GPT-4V.

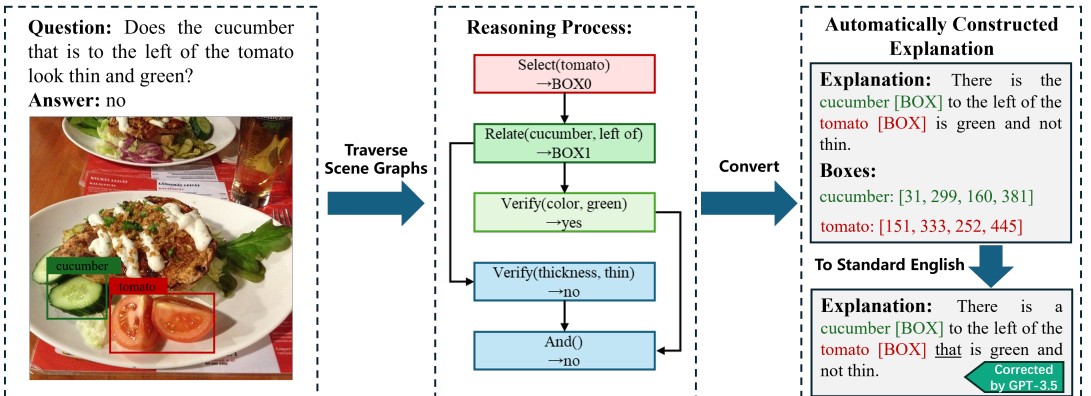

**Figure 3: Brief construction procedure of our multimodal explanations: (1) Traverse scene graphs to link reasoning processes with visual objects; (2) Convert reasoning processes into multimodal explanations by automatic programs; (3) Convert the text (excluding [*BOX*]) into standard English by GPT-3.5 and add [*BOX*] back.**

## 2 RELATED WORK

We review three highly related topics in explainable Visual Question Answering (VQA).

**Textual explanation for VQA.** Researchers have long recognized the importance of explanations in yielding verifiable reasoning results and establishing trustworthy VQA systems. Early attempts focus on providing textual explanations [21, 22, 28, 30, 44]. Zhou et al [53] propose a multi-task model to predict the answer to a visual question and generate the image caption to explain the answer simultaneously. Since their VQA data and image captioning data are from independent sources, the caption may be irrelevant to the question. Following their work, Li et al. [22] propose the VQA-E dataset by adopting the answer-related caption as the explanation. Differently, Wen and Peng [47] propose to retrieve related text from databases to explain the reasoning process.

Though textual explanations are simple and easy to obtain, researchers have also acknowledged their limitations. While VQA involves reasoning on multiple modalities, textual explanations often fail to effectively explain the visual concepts involved in the reasoning processes. Therefore, recent research has begun to explore Multimodal Explanation for VQA (MEVQA).

**Multimodal explanation for VQA.** Early research on MEVQA attempts to provide individual visual explanations in addition to textual explanations [34, 48]. These works typically relate visual attention maps to the predicted answers or explanations. However, attention maps can be confusing. For example, textual explanations "a man riding on a horse" and "a horse riding on a man" can have similar attention maps. To provide more expressive and accurate explanations of the reasoning processes, joint language-vision explanations are proposed [7, 50, 51]. For example, Chen and Zhao [7] propose to represent visual objects in the textual explanations by their grounding number predicted by a trained Faster R-CNN [36], belonging to only 81 classes. Moreover, their constructed explanations are in unnatural format, reducing the readability and applicability. Following their research, Xue et al. [50] recently propose a variational causal model to improve answer-explanation consistency by establishing the corresponding causal correlation.

However, traditional MEVQA methods rely on large-scale training data, which are costly and may be impractical in various vertical domains. Moreover, considering that the recent open-world models (e.g., LLMs) have already learned open-world abilities [27, 46, 52], more streamlined and applicable learning frameworks may represent the future direction. Therefore, we propose the SME dataset, where the explanations are in standard English with additional visual grounding tokens, supporting the research of Few-Shot MEVQA (FS-MEVQA). To perform FS-MEVQA, our proposed MEAgent can leverage a GPT-3.5-based agent with multimodal open-world tools to infer answers and generate multimodal explanations for visual questions, given only $N(= 16)$ training samples.

**Few-shot explanation for VQA** Recent studies have found that LLMs are powerful few-shot learners [6, 23, 38]. Therefore, several LLM-based few-shot explanation methods have been developed [28, 29], which mainly focus on textual explanation. For example, Lu et al. [28] input the image caption and the question with in-context examples into GPT-3 to generate a chain-of-thought (i.e., textual explanation) and the answer. Different from the existing methods, we propose a few-shot method to generate multimodal explanations for visual questions, thereby enhancing the understanding of multimodal reasoning processes for VQA.

## 3 DATASET

Explaining reasoning processes for visual questions can aid in understanding and verifying the predicted answers, thereby enhancing the reliability and credibility of VQA systems. Our proposed dataset aims to provide an effective benchmark for MEVQA. Compared to previous multimodal explanations for VQA [7, 34, 51], our dataset has three key advantages: (1) Instead of separate textual and visual explanations [34], we integrate language and vision to construct brief but expressive explanations; (2) Instead of using unnatural formats [7, 51], our explanations are in standard English with additional [*BOX*] tokens for visual grounding, bridging to broad advancements in Natural Language Processing (NLP). We annotate both object boxes and their corresponding names mentioned in explanations, which support (3) a more effective visual metric for measuring the ability to simultaneously generate names and ground regions of key visual objects to explain the multimodal reasoning processes. Figure 3 demonstrates a brief construction procedure of our multimodal explanations.

**Data collection.** We leverage VQA samples in the GQA dataset [15]. Given the 127 different operations in GQA, we first represent each reasoning step in the reasoning process by a <operation, relation/attribution, dependency 1, dependency 2> tuple, where "dependency 1" and "dependency 2" can be previous reasoning steps. Then, we follow [7] to categorize all operations into 12 atomic operations that cover the essential semantics, i.e., *Select*, *Exist*, *Filter*, *Query*, *Verify*, *Common*, *Same*, *Different*, *Compare*, *Relate*, *And*, and *Or*. For example, to answer "Does the cucumber that is to the left of the tomato look thin and green?", we need to find the tomato first by executing "*<Select, tomato, None, None>*", or "*Select(tomato)*" in abbreviation.

### 3.1 Multimodal Explanation Construction

With reasoning step tuples, we need to convert reasoning steps into explanations. Previous work [7] uses a Faster-RCNN [36] trained on MS-COCO [25] to annotate visual objects related to reasoning steps, since their proposed explanation model also uses Faster-RCNN to extract visual objects. However, Faster-RCNN is often inaccurate for annotating objects in the GQA dataset. In contrast, we traverse the human-annotated scene graphs to obtain the results of all steps and link reasoning steps with their related visual objects. Then, we convert the reasoning process graphs into explanations with templates designed for all operations. For example, a *Verify* operation "*<Verify, ATT, DEP1, None>*" is converted into "(DEP1) is [RETURN] ATT", where [RETURN] is "not" if this operation returns "no" or empty otherwise. (DEP1) is the phrase converted from the dependent operation. Moreover, we use the [BOX] tokens in explanations to represent the grounding boxes of visual objects related to the reasoning steps. To facilitate the visual metric presented in Section 3.2, we annotate both the values of grounding boxes and their corresponding names mentioned in the explanations. We also design programs to correct some grammar issues in generated explanations, e.g., merging two *Verify* operations in Figure 3. However, rigid programs cannot address all language errors in the constructed explanations. Therefore, we leverage the powerful language ability of GPT-3.5 [6] to correct our explanations. To avoid disruptions of [BOX], we remove [BOX] in the explanations and utilize GPT-3.5 to convert the explanations into standard English. Then, we add back the [BOX] tokens according to their names. Finally, we take several rounds of manual checks to further clean the constructed explanations. After removing a small number of low-quality samples, we obtain 1,028,230 multimodal explanations in standard English with additional [BOX] tokens for visual grounding. All data are divided into 901,203 training samples, 97,027 validation samples, and 30,000 test samples. In the MEVQA task, the model is required to answer the visual question, generate a textual representation of the explanation (e.g., "There is a cucumber [BOX] to the left of the tomato [BOX] that is green and not thin."), and grounds visual boxes linked to all [BOX] tokens.

### 3.2 Metrics

To evaluate the generated multimodal explanations (aka., reference explanations), we adopt textual and visual metrics. **For textual metrics**, since we only add [BOX] into standard English, we directly adopt widely-used language metrics, i.e., BLEU-4 [33], METEOR [5], ROUGE-L[24], CIDEr [43], and SPICE [2]. **For visual**

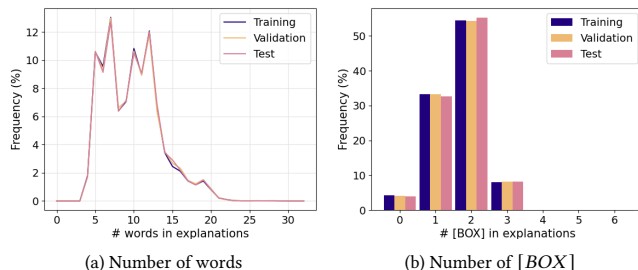

(a) Number of words      (b) Number of [BOX]

**Figure 4: Distributions of the explanation lengths and the numbers of [BOX] in explanations of our SME dataset.**

**metrics**, we propose a new detection metric to comprehensively evaluate the ability to generate the names of visual objects and ground their corresponding regions in images. For every object name $s$ annotated in a ground truth explanation (e.g., "cucumber" and "tomato" in Figure 3), we match the [BOX] token following $s$ in the reference explanation. Then, we compute the IoU (intersection over union) score between the ground truth boxes $B_{gt}^s$ of $s$ and the reference boxes $B_{re}^s$ related to this [BOX] token, evaluating the detection precision of this object. The final detection score of one explanation is averaged over all object names, as follows:

$$Detection = \frac{1}{N} \sum_s IoU(B_{gt}^s, B_{re}^s), \qquad (1)$$

where $N$ is the number of object names that occur in the ground truth and reference explanations. Therefore, the redundant boxes in the reference explanation and the missing boxes can punish the final detection score. Moreover, we propose visual attribution metrics to evaluate the ability of models to understand key visual attributions and generate them in explanations for MEVQA, which is introduced in Section 5.5.

More discussions of our adopted metrics, especially their improvements compared to previous work, are included in Supplementary Materials.

### 3.3 Data Analysis

Our SME dataset is composed of 901,203 training samples, 97,027 validation samples, and 30,000 test samples.

**Distribution of Multimodal Explanations.** The main statistics of multimodal explanations in the training, validation, and test sets are shown in Figure 4. Three split sets have similar distributions of the explanation length and the number of [BOX] in explanations. The explanation lengths range from 3 words to 32 words. The majority (97.0%) of explanations consist of between 5 to 19 words, indicating the requirement of generating brief but expressive text. The numbers of [BOX] in explanations range from 0 to 6, which corresponds to the number of visual objects that should be grounded for explaining. The majority (95.8%) of explanations contain at least 1 [BOX], highlighting the crucial role of vision in our multimodal explanations. When comparing three split sets, the test set shows a slightly higher proportion of explanations with 2 or 3 [BOX], which improves the difficulty of the test set.

More information about our dataset is provided in Supplementary Materials.

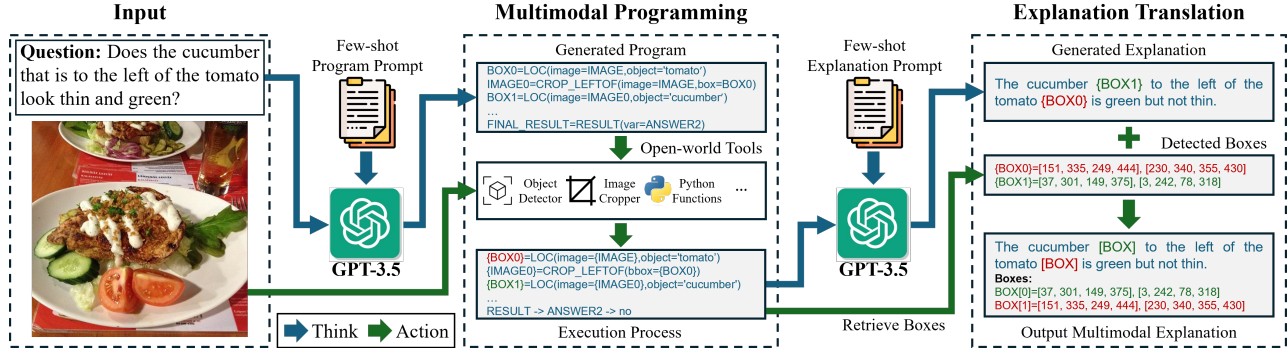

**Figure 5: Framework of our proposed Multimodal Explaining Agent (MEAgent) method: (1) Convert the input question into the program and execute it by open-world tools, with a few-shot program prompt; (2) Translate the execution process into the explanation with a few-shot explanation prompt and link the detected visual boxes to form the multimodal explanation.**

## 4 METHOD

State-of-the-art MEVQA methods [7, 34, 50] rely on large-scale training data for learning explanation, despite being built upon pretrained Vision-Language Models (VLMs). However, annotating VQA data with explanations is costly and may be impractical in various vertical domains. Meanwhile, recent research [14, 29, 40] has shown that LLM-based visual programming is effective for few-shot VQA. Inspired by the above observations, we propose a training-free Multimodal Explaining Agent (MEAgent) method for Few-Shot MEVQA (FS-MEVQA) in this paper, given merely $N(= 16)$ training samples. MEAgent leverages GPT-3.5 and multimodal open-world tools to conduct think and actions for FS-MEVQA, composed of Multimodal Programming and Explanation Translation stages. Figure 5 illustrates the framework of our method.

**Multimodal Programming.** In Multimodal Programming (MulProg), we decompose the input question $Q$ into program steps, which are then executed with the input image $I$ by multimodal open-world tools. Specifically, we select a set of multimodal open-world tools as our program modules. For example, we use the open-world object detection model OWL-ViT [31] as our location module $LOC(image, object)$. These modules can be flexibly customized and expanded to improve the reasoning capability of our MEAgent. To convert $Q$ into a program instantiated by pre-defined program modules, we use $N(= 16)$ in-context examples to form the few-shot program prompt that exemplifies the correspondence between questions and programs. To improve the effectiveness, our designed prompt includes all 16 selected modules. By inputting the prompt and the input question $Q$, GPT-3.5 can comprehend the correspondence and generate the program for solving $Q$. Then, multimodal open-world tools are activated to execute the program, inferring the answer with a program-like execution process. We denote the textual representation of the execution process as $P$, while values of variables are not directly contained in $P$. More details of Multimodal Programming are included in Supplementary Materials.

**Explanation Translation.** Though the MulProg stage provides a program-like execution process for solving the visual question, such a process is abstract, verbose, and hard to read by general users. Therefore, this stage aims to translate the execution process into a user-friendly multimodal explanation. The textual representation $P$ of the execution process obtained by MulProg can be comprehended by GPT-3.5. Motivated by this, as shown in Figure

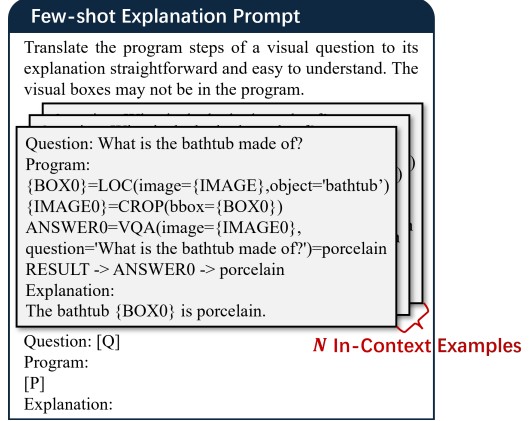

**Figure 6: Our few-shot explanation prompt. [Q] denotes the input question and [P] denotes its execution process.**

6, we construct a few-shot explanation prompt, including $N(= 16)$ in-context examples to demonstrate the correspondence between execution processes and explanations in standard English with additional $\{BOXi\}$ tokens that represent the box variables in programs. For the input question $Q$ and its execution process $P$, the prompted GPT-3.5 can generate the explanation for the reasoning process. However, since MulProg aims to solve the visual question and neglects to explain the reasoning process, some key visual objects for explanation may be ignored in the generated program. To address this problem, we propose a simple yet effective rethinking mechanism to improve the generated explanation. As shown in our prompt, MEAgent allows GPT-3.5 to directly output the needed objects that may not be in the program. Next, MEAgent checks all box variables in the generated explanation and links those existing in the program to their computed values. For visual boxes ignored in the program, MEAgent extracts their object names in the explanation and utilizes the location module $LOC$ to compute their values. Finally, MEAgent can obtain the multimodal explanation with $[BOX]$ tokens to represent boxes of key visual objects.

## 5 EXPERIMENTS

We conduct extensive experiments on the proposed SME dataset to investigate the effectiveness of our proposed MEAgent method for FS-MEVQA. More details are included in Supplementary Materials.

**Table 2: The results of Few-Shot Multimodal Explanation for Visual Question Answering on the SME dataset. $N$ denotes the number of training samples. Gray results are trained on the whole training set.**

| Method | Backbone | $N$ | BLEU-4 | METEOR | ROUGE-L | CIDEr | SPICE | Detection | ACC |
|--------|----------|-----|--------|--------|---------|-------|-------|-----------|-----|
| REX | VisualBERT | 16 | 0.00 | 4.37 | 23.23 | 0.89 | 0.00 | 0.00 | 17.77 |
| REX | VisualBERT | 64 | 0.00 | 4.84 | 24.08 | 0.91 | 0.00 | 0.00 | 17.77 |
| REX | VisualBERT | 256 | 0.00 | 4.87 | 24.76 | 0.98 | 0.00 | 0.00 | 18.05 |
| REX | VisualBERT | 1K | 8.84 | 10.83 | 38.60 | 34.86 | 4.10 | 0.43 | 18.19 |
| REX | VisualBERT | 4K | 33.88 | 24.98 | 59.24 | 99.31 | 26.63 | 1.84 | 27.50 |
| REX | VisualBERT | 16K | 56.35 | 39.22 | 75.49 | 275.16 | 48.42 | 3.94 | 33.04 |
| REX | VisualBERT | All | 84.26 | 56.22 | 90.50 | 763.63 | 78.88 | 7.66 | 61.38 |
| REX | LXMERT | All | 87.45 | 58.90 | 92.15 | 798.55 | 82.69 | 7.82 | 71.91 |
| VCIN | VisualBERT | 16 | 9.17 | 19.82 | 33.34 | 4.28 | 13.39 | 0.28 | 17.77 |
| VCIN | VisualBERT | 64 | 20.27 | 25.61 | 47.53 | 9.72 | 25.97 | 0.41 | 17.77 |
| VCIN | VisualBERT | 256 | 29.69 | 28.94 | 53.96 | 24.35 | 27.96 | 0.67 | 18.05 |
| VCIN | VisualBERT | 1K | 37.00 | 31.99 | 59.07 | 54.01 | 29.84 | 1.10 | 24.10 |
| VCIN | VisualBERT | 4K | 47.07 | 35.49 | 67.91 | 142.59 | 39.65 | 4.79 | 28.62 |
| VCIN | VisualBERT | 16K | 59.42 | 41.54 | 75.54 | 309.85 | 50.93 | 9.77 | 35.71 |
| VCIN | VisualBERT | All | 90.64 | 61.80 | 93.73 | 833.37 | 86.63 | 21.90 | 64.00 |
| VCIN | LXMERT | All | 91.52 | 62.96 | 94.44 | 847.43 | 88.41 | 23.04 | 73.31 |
| GPT-4V | GPT-4V | 16 | 45.51 | 35.17 | 52.67 | 269.68 | 37.67 | 7.00 | 42.30 |
| **MEAgent (Ours)** | **GPT-3.5** | **16** | **67.91** | **50.55** | **79.41** | **510.44** | **64.09** | **29.09** | **51.45** |

## 5.1 Experimental Setup

**Baseline methods.** We adopt three baseline methods, including two state-of-the-art methods for MEVQA, as follows:

**REX** [7] grounds visual objects by Faster-RCNN [36] and utilizes an LSTM-based generator to generate multimodal explanations, based on features extracted by pretrained Vision-Language Models (VLMs).

**VCIN** [50] is the state-of-the-art method for MEVQA, which utilizes a gating Transformer to generate explanations and establish causal correlations to improve explanation-answer consistency, based on Faster-RCNN and pretrained VLMs.

Experiments reported in [50] adopt VisualBERT [20] and LXMERT [41] as backbone VLMs. Given that LXMERT is pretrained on the GQA dataset which overlaps with the questions and images in our dataset, we only adopt VisualBERT in few-shot experiments.

**GPT-4V(ision)** [1] may be currently the most powerful multimodal LLM. Since GPT-4V can generate text and detect visual objects, we also construct a prompt with the same $N(= 16)$ examples in our method to facilitate question answering and multimodal explanation generation via GPT-4V. The details are included in Supplementary Materials.

**Evaluation.** As introduced in Section 3, we adopt BLEU-4 [33], METEOR [5], ROUGE-L[24], CIDEr [43], and SPICE [2] to evaluate the language quality. To evaluate the ability to generate the names of visual objects and ground their corresponding regions in images, we leverage our proposed visual detection metric. Moreover, we report the answering accuracy (abbr., ACC).

**Implementation details.** While our SME dataset of 1,028,230 samples can also be adopted in the research of traditional MEVQA, we focus on FS-MEVQA in our experiments. In the few-shot setting, different from traditional $N$-shot learning [8, 39, 45], we randomly sample $N$ training samples for training and evaluate models on the whole test set. All results are averaged on 5 runs with different random seeds. More details are in Supplementary Materials.

## 5.2 Results and Discussions

The experimental results on our SME dataset are shown in Table 2. From the results, we have the following observations:

- The performance of traditional MEVQA methods (i.e., REX and VCIN) significantly drops when $N$ is small. This shows these methods cannot effectively conduct FS-MEVQA, though being built upon pretrained VLMs. These methods rely on large-scale training data, which may not be an ideal training paradigm in practice, especially considering the recent progress in open-world learning and LLM.

- The detection scores of REX and VCIN are much lower than that of our MEAgent, even trained on all 901,203 training samples. This can be attributed to their adopted Faster R-CNN for pre-extracting 36 objects for every image, which may not contain the needed objects and lacks open-world objects. Differently, we use an open-world detector OWL-ViT to detect objects with their names mentioned in explanations.

- Our MEAgent significantly outperforms GPT-4V with the same $N$ and state-of-the-art MEVQA methods with even thousands of training samples, showing its effectiveness for FS-MEVQA. Though 16 samples are insufficient for learning question answering and explaining, they are effective in defining our standard English-like multimodal explanations. Based on this, our LLM agent-based method can leverage out-of-training knowledge and multimodal open-world tools to perform FS-MEVQA.

- The language quality (i.e., BLEU-4, METEOR, ROUGE-L, CIDEr, and SPICE) of REX and VCIN trained on all training samples significantly outperforms GPT-4V and our MEAgent with few training samples. This shows the quality of our constructed training set and the current performance gap between traditional MEVQA and FS-MEVQA, suggesting more future work on FS-MEVQA.

**Figure 7: Qualitative results of the generated multimodal explanations and the predicted answers. The [BOX] tokens link to visual boxes of the same colors in the images.**

**Table 3: Ablation results on the SME test set. $N$ denotes the number of training samples. MEAgent-RTK denotes MEAgent without the rethinking mechanism.**

| Method | $N$ | BLEU-4 | METEOR | ROUGE-L | CIDEr | SPICE | Detection |
|--------|-----|--------|--------|---------|-------|-------|-----------|
| MEAgent-RTK | 16 | 65.15 | 46.31 | 75.66 | 456.03 | 55.39 | 23.58 |
| MEAgent | 4 | 52.50 | 41.72 | 69.61 | 372.97 | 45.32 | 15.77 |
| MEAgent | 8 | 62.93 | 47.39 | 75.85 | 477.38 | 59.06 | 22.52 |
| **MEAgent** | **16** | **67.91** | **50.55** | **79.41** | **510.44** | **64.09** | **29.09** |

## 5.3 Ablation Study

In this section, we ablate our rethinking mechanism and investigate the effect of $N$ in our method. The ablation results are shown in Table 3, from which we have the following observations: (1) MEAgent-RTK removes the rethinking mechanism and only uses visual objects detected in multimodal programming, which performs worse than MEAgent, especially on the detection metric. This indicates the effectiveness of the rethinking mechanism which can generate and ground objects ignored in the previous multimodal programming. (2) MEAgent performs better with more training samples (i.e., $N$). However, compared to the results in Table 2, our MEAgent with $N = 4$ can still outperform GPT-4V with $N = 16$ or REX and VCIN with $N = 4$K. Our MEAgent with $N = 8$ can still outperform REX and VCIN with $N = 16$K. These results further verify the effectiveness of our method.

## 5.4 Qualitative Results

In addition to quantitative results, we demonstrate qualitative results of the predicted answers and explanations for FS-MEVQA

in Figure 7. Compared to VCIN ($N$=16K) and GPT-4V ($N$=16), our MEAgent ($N$=16) can generate more rational, accurate, and coherent explanations of the reasoning process: (1) In (b)-(d), VCIN cannot ground the key visual objects in the images for explanation, while our MEAgent can utilize the names of objects and an openworld detector to locate accurate boxes. (2) VCIN usually predicts inconsistent answers and explanations. This is because VCIN needs large-scale data to learn the causal correlation between the answer and explanation. Differently, we utilize LLM to translate the execution process of inferring the answer to its explanation, ensuring inherent consistency between answers and explanations. (3) In (b)-(d), the detection accuracy of GPT-4V is also unsatisfactory. In (d), GPT-4V grounds an unimportant object (i.e., "wall") in the explanation but ignores key objects (i.e., "soap dispensers" and "bathroom"), which shows GPT-4V fails to capture the correct reasoning process for solving this question. Differently, our MEAgent explicitly generates the multimodal program for solving the input question and generates the explanation accordingly. Therefore, MEAgent can capture rational reasoning processes more effectively.

## 5.5 Results of Visual Attribution Metrics

We further propose metrics to evaluate the ability of models to understand key visual attributions and generate them in explanations for FS-MEVQA. We have collected 38, 41, 16, 96, 6, 11, 14, 9, 79, 38, and 43 keywords about colors, materials, shapes, activities, sizes, poses, sports, directions, animals, people, and plants that occur in the test explanations, separately. Then, we compute the percentage of correctly generated keywords in the explanations for every

**Table 4: The accuracy of explaining key visual attributions on the SME test set. *N* denotes the number of training samples.**

| Method | *N* | Color | Material | Shape | Activity | Size | Pose | Sport | Direction | Animal | Person | Plant |
|--------|-----|-------|----------|-------|----------|------|------|-------|-----------|--------|--------|-------|
| VCIN | 16K | 31.65 | 24.45 | 10.50 | 59.76 | 39.21 | 33.40 | 16.29 | 55.23 | 29.79 | 70.39 | 31.07 |
| GPT-4V | 16 | 63.11 | 59.07 | 61.43 | 65.27 | 70.48 | 71.56 | 68.42 | 69.13 | 72.73 | 48.68 | 60.00 |
| **MEAgent** | **16** | **82.67** | **76.90** | **83.60** | **95.25** | **79.77** | **92.46** | **89.71** | **87.92** | **82.97** | **83.45** | **67.85** |

**Question:** Are there any injured fingers?

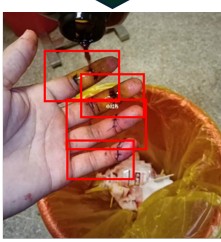

**Explanation:** There are 4 injured fingers [BOX].
**Answer:** yes

**(a) Healthcare**

**Question:** What color is the smoke above the factories?

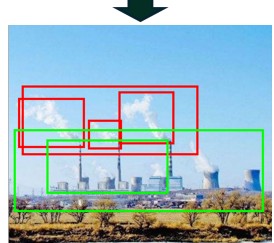

**Explanation:** The smoke [BOX] above the factories [BOX] is white.
**Answer:** white

**(b) Industry**

**Question:** Is the bigger celestial object a star or a planet?

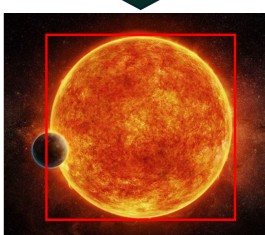

**Explanation:** The bigger celestial object [BOX] is a star.
**Answer:** star

**(c) Science**

**Question:** On which side of this chart is the title?

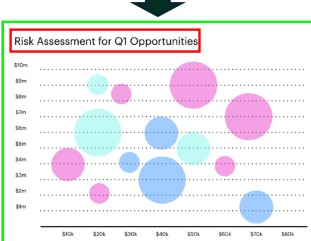

**Explanation:** The title [BOX] of the chart [BOX] is located on the left side.
**Answer:** left

**(d) Office**

**Figure 8: Out-of-distribution results of our MEAgent for FS-MEVQA. The [BOX] tokens link to visual boxes of the same colors in the images.**

attribution, which are demonstrated in Table 4. From the results, we have the following observations: (1) Explaining plants appears to be the most challenging one, which may be due to the limited training data on subdivided plant species in current research. Therefore, the experimented LLMs and VLMs may have less knowledge about plant species. (2) Interestingly, GPT-4V demonstrates superior performance compared to VCIN across all attributions except for *Person*, despite having lower language scores in Table 2. This can be attributed to the extensive open-world knowledge of GPT-4V, while VLM-based VCIN suffers from insufficient knowledge in few-shot learning. This comparison also shows the complementarity of visual attribution metrics in evaluating visual knowledge for explanation. (3) GPT-4V achieves an especially low score for *Person*, which indicates GPT-4V may be too cautious in identifying the biological sexes, ages, and occupations of people. (4) Our MEAgent significantly outperforms both baselines, showing its superior ability to understand visual attributions and explain them. Though GPT-3.5 can only understand and generate text, the GPT-3.5-based MEAgent can leverage open-world visual tools (e.g., object detector and image croppers) to perform multimodal explanation, even outperforming the stronger multimodal LLM GPT-4V.

## 5.6 Out-Of-Distribution Results

Given the limited knowledge within training samples in FS-MEVQA, a key to performing FS-MEVQA is leveraging open-world knowledge. Therefore, we further gather Out-Of-Distribution (OOD) visual questions that involve concepts not present in our SME dataset to test our MEAgent method. In Figure 8, we show the results for four OOD samples belonging to healthcare, industry, science, and office domains. We are surprised to find that MEAgent can predict rational answers and generate good explanations for these questions. In (a), since we implement a *COUNT* module to count the number of detected boxes, MEAgent can even explain the number of injured fingers. In (d), while MEAgent can accurately locate the title and the chart, its answer is not very accurate ("upper left" may be a better answer). These results may suggest a boarder application range of our MEAgent in various domains. By employing an LLM agent with open-world tools for visual reasoning and multimodal explanation, MEAgent is capable of overcoming the limited knowledge in training data and generalizing to encompass open-world questions. Moreover, we argue that utilizing domain-specific programming modules and examples can further improve the capability of MEAgent in a particular domain.

## 6 DISCUSSION AND CONCLUSION

In this paper, we propose a Standard Multimodal Explanation (SME) dataset with 1,028,230 samples for Visual Question Answering (VQA) with elaborately constructed multimodal explanations of the underlying multimodal reasoning processes. Based on our dataset, we propose a new Few-Shot Multimodal Explanation for VQA (FS-MEVQA) task, which aims to answer the visual question and explain the reasoning process with a limited number (denoted as *N*) of training samples. To the best of our knowledge, SME is the first large-scale dataset for MEVQA with joint language-vision explanations based on standard English and additional visual grounding tokens, which bridge MEVQA to a broad field in Natural Language Processing (NLP). Moreover, we propose a training-free Multimodal Explaining Agent (MEAgent) method based on an LLM agent with multimodal open-world tools for FS-MEVQA. Extensive experiments demonstrate that our MEAgent significantly outperforms traditional MEVQA methods and GPT-4V for FS-MEVQA.

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
