# OpenReview forum: "Few-Shot Multimodal Explanation for Visual Question Answering"
_acmmm.org/ACMMM/2024/Conference — MM2024 Poster_

### Official Review · Reviewer_7nw3 · 2024-05-24

**Rating:** 3
**Confidence:** 3

**Summary:**

This paper proposes a large-scale VQA dataset for MEVQA tasks and proposes a training-free MEAgent framework to solve the MEVQA tasks. This method has better results on the MEVQA data set than comparison methods.

**Strengths:**

1. The author proposes a novel MEVQA dataset(SME) based on GQA, whose quantity and annotation are improved compared with existing datasets.
2. Compared with other MEVQA methods, the method proposed by the author has better results in terms of accuracy and interpretability indicators.

**Limitations:**

1. The description of GPT-4V and MEAgent in Figure 2 is inaccurate. For GPT-4V and MEAgent, should it be few-shot samples instead of training samples? This problem appears many times elsewhere in the paper (eg, line 223, table 3). However, based on the descriptions of line 186 and line 202, it can be understood that the authors meant the number of in-context examples. I suggest the author modify the description for GPT-4V and MEAgent.
2. The motivation behind this work is somewhat confusing. In my view, this work is based on visual programming. That is, based on visual programming, LLM is used to combine the detected boxes to give interpretability in standard English. But visual programming itself already has strong interpretability and clear intermediate processes. I noticed the author's explanation in line 517, that is, the code is abstract and difficult to understand for users, but the execution process of the code does not only include the code, such as the example on the VperGPT homepage (https://viper.cs.columbia.edu/), is visual programming with intermediate execution processes and results inherently more interpretable than textual language? Therefore, from this perspective, such a conversion from code to standard English is somewhat weak.
3. Will the SME data set based on the [BOX] tag limit its diversity of explanations about the question? For example, in Figure 8(c), it seems that [BOX] does not play a role in explaining the answer.

**Suitability:**

3

---

### Official Review · Reviewer_gvXZ · 2024-05-26

**Rating:** 3
**Confidence:** 2

**Summary:**

The paper introduces a new dataset Standard Multimodal Explanation and the task of Few-Shot Multimodal Explanation for VQA. The SME dataset consists of over a million samples that include multimodal explanations. The authors propose a novel method, the Multimodal Explaining Agent , using a large language model (LLM) combined with multimodal open-world tools to generate explanations from a minimal number of training samples. The experimental results demonstrate that MEAgent significantly outperforms existing MEVQA methods and the multimodal LLM GPT-4V, particularly in few-shot learning scenarios. This work contributes to advancing explainable AI by improving the readability and usability of explanations in VQA systems.

**Strengths:**

1. This paper is integral in general and able to justify itself, fulfilling what it takes to propose a new dataset and task.

2. The proposed method is effective on the proposed dataset, making the combination GPT3.5 and off-the-shelf tools more effective than GPT4V.

3. I believe the generated multimodal explanation is indeed more interpretable and fluent than previous works.

**Limitations:**

1. I doubt the necessity in the proposal of the task. Is it really necessary to combine multimodal explanation with few-shot scenarios? It is relatively easy to find multimodal rationales even for low-resourced languages as long as there are image-text pairs, especially in the current era when MLLMs are good at generating such information.

2. In results comparison, only gpts and ancient methods like LXMERT and VisualBERT are present, which were proposed five years ago. I expect more results from recent MLLMs-based methods.

3. Using formulaic metrics like BLEU and CIDER might not bring fair evaluation. The generation styles of models vary, and results from GPT4V might be hard to hit the expected text.

(Minor) Clerical error: In line 922, bridge->brigdes

**Suitability:**

3

---

### Official Review · Reviewer_CLZa · 2024-06-07

**Rating:** 4
**Confidence:** 2

**Summary:**

The paper focuses on advancing explainable artificial intelligence (XAI) in the context of Visual Question Answering (VQA) systems. It introduces two main contributions: a new dataset called Standard Multimodal Explanation (SME) dataset and a method named Multimodal Explaining Agent (MEAgent) for Few-Shot Multimodal Explanation for VQA (FS-MEVQA). The SME dataset consists of over a million samples containing questions, images, answers, and multimodal explanations, enabling research in both traditional MEVQA and FS-MEVQA. The MEAgent method employs a training-free approach based on a LLM agent to generate multimodal explanations and infer answers for visual questions, particularly with few training samples.

**Strengths:**

1.New dataset: The paper introduces a new dataset, SME, which is substantial in scale and includes joint language-vision explanations with visual grounding tokens. The proposed dataset could be utilized in the field of interpretable VQA tasks, both for training and evaluating.

2.Comprehensive Evaluation: The paper presents comprehensive experimental results, evaluating the proposed method using various metrics including language quality, visual detection, and visual attribution. This thorough evaluation demonstrates the effectiveness and superiority of the proposed approach over existing methods.

**Limitations:**

I believe the main contribution of the paper lies in the proposed SME dataset. As for the proposed MEAgent framework, I'm concerned about its innovativeness, considering that using LLM as an agent to access open-world tools has been widely employed in previous works(for example, GPT4Tools: Teaching Large Language Model to Use Tools via Self-instruction). The implementation of few-shot training using the In-context learning method is also not novel in this work. However, considering the novelty of the task in providing explanations with bounding box annotations when answering questions, I would still give a borderline accept result. Nevertheless, I think the experimental section of this work could be supplemented. I believe comparisons with methods trained on large amounts of data are unnecessary. Instead, experiments should be expanded to include: 1) experiments on more multimodal large language models; 2) experiments on LLM-based agents and other related models to validate the effectiveness of this work.

**Suitability:**

3

---

### Official Review · Reviewer_QNM9 · 2024-06-08

**Rating:** 4
**Confidence:** 3

**Summary:**

This paper proposes a standard multi -mode data set SME, which contains special labels to associate with visual information. Given the latest progress in open-world learning, the author proposes a stronger and more practical small sample learning architecture, which includes an LLM agent based on GPT-3.5. The agent transformed the execution process into a multi-mode explanation of the multi-mode reasoning process through an open tool reasoning. In addition, the paper also considers the effects of visual objects that are often overlooked in multi-model training.

**Strengths:**

1. The author uses the latest external research progress as an agent to assist in conducting visual Q & A multi-modal interpretation research. This is an exciting and effective method. This saves research costs and achieves ideal results in a shorter time.
2. The SME dataset is extensive, with over a million samples, and is the first to provide joint language-vision explanations in standard English with visual grounding tokens. The new architecture dataset is very helpful for subsequent research. Also, the dataset bridges MEVQA to NLP, making it relevant for broader applications.
3. The MEAgent method is training-free and can generate multimodal explanations from only 16 training samples. It utilizes open-world tools and an LLM agent, showing significant performance improvements over state-of-the-art methods.
4. The experimental setup is robust, including language quality metrics, visual detection, and visual attribution metrics.

**Limitations:**

1. Although the experimental results show significant improvement in reasoning ability, it is unclear whether this enhancement is solely due to the combination of strong models (LLMs ) or other factors.
2. The authors' concept of few-shot visual question-answer learning may not be entirely accurate. While the number of training samples is small, the foundational components, such as object detection models and LLM agents, have been pre-trained on extensive datasets.
3. In the Dataset section, the authors describe the reasoning process using tuples to represent reasoning steps. However, this method may ignore the complex dependencies between steps, not just two simple dependencies.
4. The paper lacks detailed case studies or examples of real-world applications where the proposed method has been implemented successfully.
5. The proposed method, it is innovative but may be complex to implement and require significant computational resources for execution, especially when integrating various multimodal tools.
6. The reliance on pre-trained models, such as GPT-3.5, introduces the risk of biases present in these models, which could affect the fairness or accuracy of the explanations generated by the system and may cause hallucination problem.

**Suitability:**

3

---

### Official Review · Reviewer_yDuB · 2024-06-08

**Rating:** 4
**Confidence:** 3

**Summary:**

This paper proposed the Standard Multimodal Explanation (SME) dataset and the Multimodal Explaining Agent (MEAgent), a novel approach to enhance explainable visual question answering (VQA) through few-shot learning.
The SME dataset include a wide range of human-annotated visual objects linked with structured language expressions, facilitating multimodal explanations. MEAgent leverages a pre-trained Large Language Model (LLM) to interpret and generate explanations for VQA tasks using only a small subset of examples from the SME dataset. This approach enables the MEAgent to deliver high quality multimodal explanations with minimal training, outperforming existing state-of-the-art methods in both accuracy and the robustness of generated explanations. The paper also includes a innovative visual evaluation metrics that can penalise missing and redundant elements in explanations, ensuring precise visual grounding and enhancing the model's performance in practical scenarios.

**Strengths:**

1. The SME dataset uses human-annotated classes which provide a more comprehensive and varied set of classes for the model to learn from. It also features more structured language expressions which are user-friendly compared to earlier datasets.
2. The proposed method introduces a novel visual evaluation metric for measuring the detection score, allowing missing and redundant boxes to impact the detection score negatively.
3. The MEAgent requires only a small subset of samples from the SME dataset to achieve state-of-the-art performance, while other frameworks might need many more samples to do so.

**Limitations:**

1. The explanation for the initialisation and generation of the few-shot program prompts seems insufficient. As this is a fundamental component of the proposed structure, a more detailed description would benefit the understanding of in-context learning.
2. In the comparisons, different backbones were tested using different numbers of samples per class, however, the complexity level or number of parameters of different backbones was not considered, which can be crucial when adapting to real-life scenarios.
3. Despite better performance in accuracy for MEAgent compared to baseline models, the processing speed of the model was not mentioned, which is crucial in terms of comparing model efficiency.
4. In the ablation study section lines 745-746, it claims that MEAgent with N=4 is able to outperform GPT-4V with N=16. However, the accuracy for MEAgent with N=4 is missing from the table, thus this conclusion lacks evidence support at this stage.

**Suitability:**

3

---

### Official Review · Reviewer_v3JF · 2024-06-08

**Rating:** 3
**Confidence:** 3

**Summary:**

This paper introduces large-scale Standard Multimodal Explanation dataset for multimodal explanation for Visual Question Answering (MEVQA).

Compared to other datasets, the proposed dataset not only exceeds others in the number of visual objects but also stands out as the first to both standardize explanations using formal English, and enrich them with additional visual grounding tokens, for better multimodal explanations.

Besides, a training-free method called MEAgent is proposed with the use of both LLM agent and multimodal open-world tools.

Moreover, several metrics are considered to estimate the detection performance of MEAgent.

Under the use of SME dataset, the overall performance of MEAgent exceeds other methods on few-shot multimodal Explanation.

**Strengths:**

+ The proposed dataset contains not only a large volume of data but also a wide variety of types, which better improves multimodal explanations across different scenarios.
+ Table 2 has made comparisons in a relatively comprehensive way to show MEAgent performing well on few shot Multimodal Explanation on the SME dataset.

**Limitations:**

Major:
- The number of datasets is not sufficient in Table 1 during the comparison of explanations for VQA.
- The efficiency is not taken into account in metrics for evaluating the performance of different methods.
- There lack testing results of MEAgent and other methods on other datasets apart from SME dataset to vertify the generalization ability of MEAgent.
-  The ablation study should be conducted more comprehensively, rather than just focusing on the rethinking mechanism and the effect of N.
-  The number of different methods is not sufficient in demonstrating accuracy of explaining key visual attributions.

Minor:
- explain more clearly how the few-shot program prompts interact with the backbone to enhance multimodal programming in Figure 5.

**Suitability:**

2

---

### Meta-Review · Area_Chair_kU8N · 2024-06-26

**Recommendation:** Accept (Poster)
**Confidence:** 4

**Metareview:**

This paper was reviewed by six experts, with initial ratings of borderline reject from three reviewers and borderline accept from the other three.

After the rebuttal, two reviewers increased their scores, and three maintained their original ratings. This resulted in one weak accept, two borderline accepts, and two borderline rejects (one reviewer did not finalize their rating).

Specifically, one reviewer increased their rating from borderline accept to weak accept, and another increased their score from borderline reject to borderline accept.

Given the diverse ratings for this paper, the AC carefully reviewed the paper, the reviews, and the rebuttal.

The AC identified several strengths in this paper:

(i) The new dataset is large-scale with a wide range of types, potentially contributing to multimodal explanations.

(ii) The proposed approach requires only a small subset of samples, and the newly introduced visual evaluation metric could help foster detection tasks.

(iii) The comprehensive analysis, covering visual detection, visual attribution, and language quality, demonstrates the method's effectiveness compared to existing works.

However, the AC also identified the following weaknesses:

(i) The ablation study could be improved.

(ii) The assessment of the generalization ability could be enhanced.

(iii) The motivation behind this work could be clearer.

At least two reviewers (Reviewer 7nw3 and Reviewer gvXZ) pointed out "the necessity of the proposed task".

The rebuttal addressed some concerns, but the reviewers also highlighted their remaining issues in their final justifications.

Reviewers v3JF, gvXZ, and 7nw3 outlined concerns that the AC believes would strengthen the paper. Please consider their comments in a detailed revision.

After careful consideration, the AC recommends acceptance, subject to a careful handling of the reviewers' unaddressed concerns. The AC trusts that the authors can provide a satisfactory revision.

---

### Meta-Review · Area_Chair_49sR · 2024-07-06

**Recommendation:** Accept (Poster)
**Confidence:** 4

**Metareview:**

This paper proposes to facilitate visual question answering (VQA) systems via collecting a large-scale standard multi-modal explanation dataset. This dataset is tailored for the few-shot VQA task, in which models learn to generate explanations for VQA with a few examples. A new training-free baseline is also proposed, which consists of a visual programming module and an explanation translation module for few-shot explanation generation.

The ratings of this paper were divergent, including two borderline rejects (Reviewers #v3JF and #7nw3), three borderline accepts (Reviewers #yDuB, #CLZa, and #gvXZ), and one weak accept (Reviewer #QNM9). Due to the page limitation, there were still some unsolved concerns after the rebuttal, including that (i) diverse metrics and detailed explanations regarding the prompts were needed (Reviewer #v3JF) and (ii) the necessity of the task was not clarified (Reviewers #gvXZ and #7nw3). Nevertheless, considering that related research can benefit from the proposed dataset, the AC leaned to the acceptance side. It is recommended to revise the paper carefully to address the remaining concerns and release the proposed dataset.